# Diagnostic accuracy of Savanna RVP4 (QuidelOrtho) for the detection of Influenza A virus, RSV, and SARS-CoV-2

Büsra Köse,[1] Frieder Schaumburg[1]

ABSTRACT  Seasonal increase of severe acute respiratory syndrome coronavirus 2 (SARS-CoV-2), influenza virus A/B (Flu A/B), and respiratory syncytial virus (RSV) require rapid diagnostic test methods for the management of respiratory tract infections. In this study, we compared the diagnostic accuracy of Savanna RVP4 (RVP4, QuidelOrtho) with Xpert Xpress Plus SARS-CoV-2/Flu/RSV (Xpert, Cepheid). Nasopharyngeal swabs from patients treated at a tertiary care hospital (Germany) were tested for SARS-CoV-2, Flu A/B, and RSV by RVP4 to assess diagnostic accuracy (reference standard: Xpert). The intra and inter assay precision of Ct-values was assessed by repeated test in triplicates (on day 1) and duplicates (days 2–3). All patients with a physician's order for a multiplex test for SARS-CoV-2, Flu, and RSV test were included. Duplicate swabs from the same patient, samples with a total volume ≤1 mL, or inappropriate shipment/storage were excluded. In total, 229 swabs were included between September 2023 and February 2024. The concordance between both tests was 96.5% (SARS-CoV-2), 98.7% (Flu A), and 99.6% (RSV). Flu B was not detected by both tests. The RVP4 test had a sensitivity of 85%–95% and a specificity of 100% for the detection of SARS-CoV-2, Flu A, and RSV. The intra and inter assay precision of Ct-values from RVP4 was 3% and 2% (SARS-CoV-2), 5% and 4% (Flu A), and 0% and 3% (RSV), respectively. The Savanna RVP4 has a favorable diagnostic accuracy for the detection of SARS-CoV-2, Flu A, and RSV.

IMPORTANCE  We assessed the diagnostic accuracy of a new point-of-care test for the rapid detection of severe acute respiratory syndrome coronavirus 2 (SARS-CoV-2), influenza virus A/B (Flu A/B), and respiratory syncytial virus (RSV). The new test has a concordance with the reference standard of 96.5% (SARS-CoV-2), 98.7% (Flu A), and 99.1% (RSV). The sensitivity of 85%–95% and specificity of 100% for the detection of SARS-CoV-2, Flu A, and RSV is comparable with similar nucleic acid amplification-based point of care tests but at lower costs.

KEYWORDS  Influenza A virus, Influenza B virus, respiratory syncytial virus, SARS-CoV-2, point-of-care testing, multiplex polymerase chain reaction

The incidence of respiratory tract infections caused by influenza virus A/B (Flu A/B), respiratory syncytial virus (RSV), and lately also severe acute respiratory syndrome coronavirus 2 (SARS-CoV-2) is increasing during wintertime. Influenza virus, a member of the *Orthomyxoviridae*, can be distinguished into types A, B, and C. The northern and southern hemispheres are affected by a high incidence during the cooler periods of the year (1). During one influenza season, approximately 5%–20% of Germany's population can become infected with Flu A/B, while Type A is more common than Type B (2). During epidemics, the highest infection rates are among school-aged children, while severe infections requiring hospitalization affect mostly toddlers and elderly people (1).

RSV, belonging to the *Pneumoviridae*, is widespread within all age groups but most common in infants, especially premature babies and toddlers (3). RSV season in Central

Editor Melissa R. Gitman, Icahn School of Medicine at Mount Sinai, New York, New York, USA

Address correspondence to Frieder Schaumburg, frieder.schaumburg@ukmuenster.de.

The authors declare no conflict of interest.

Europe begins mainly in November and lasts until April, while the peak occurs mostly in January and February, lasting 4–8 weeks (4).

SARS-CoV-2 is a novel Coronavirus that caused a pandemic from March 2020 to May 2023 with 772.166.517 confirmed cases worldwide (6.981.263 deaths) (5). The pandemic affected all age groups, especially elderly people and those with pre-existing conditions. Initially, SARS-CoV-2 showed high transmission rates all over the year. Lately, it can be observed that it has its peaks in the winter months (6).

Especially toddlers and elderly people, so as immunocompromised people (e.g., with cardiovascular, pulmonal or metabolic diseases, adipositas, pregnancy) are at high risk of severe or even deadly courses of infections with Flu, RSV, and SARS-CoV-2. Therefore, the importance of timely diagnosis, enabling early isolation and treatment, is key for patients' management and infection prevention and control. In addition, early confirmation of viral infection can reduce the inappropriate prescription of antibiotics in influenza-like diseases (7). Only a few platforms offer a point-of-care multiplex detection of Flu A/B, RSV, and SARS-CoV-2 in a syndromic testing approach with proven adequate accuracy. The objective of this study was, therefore, to assess the diagnostic accuracy of a novel commercial multiplex-PCR (Savanna RVP4, QuidelOrtho) using the Xpert Xpress SARS-CoV-2/Flu/RSV (Cepheid) as the reference standard.

## MATERIALS AND METHODS

### Study population

Patients were recruited from both in- and out-patient departments at the University Hospital Münster, Germany, for this prospective diagnostic accuracy study. All patients with a physician's order for a multiplex SARS-CoV-2/Flu/RSV test were included. Duplicate nasopharyngeal swab from the same patient, samples with a total volume ≤1 mL or inappropriate shipment/storage (e.g., wrong transport medium, repeated thawing, and freezing) were excluded.

The swabs (sigma-transwab, liquid amies, MWE, Corsham, Wiltshire, England) were transported to the laboratory at room temperature within 4–16 h after sampling and were stored at 6°C before testing. If tests could not be performed on the same day, the samples were stored at −80°C. Tests were run in parallel either at the time of receipt or at a later time.

### Sample size calculation

In the absence of any preliminary data on the test performance of the Savanna RVP4 plus assay, we assumed a sensitivity and specificity of 90%. Based on the assumption of a prevalence of 15% of the target virus, a precision of 10%, a 95% CI, and a drop-out rate of 15%, we calculated a total sample size of 272 samples for the study (8).

### Multiplex PCR for the detection of Flu A/B, RSV, and SARS-CoV-2

We compared the Savanna RVP4 plus (QuidelOrtho, San Diego, CA, USA) with the reference standard Xpert Xpress CoV-2/Flu/RSV plus (Cepheid, Sunnyvale, CA, USA). Both tests apply single-use disposable cartridges that include the RT-PCR reagents and provide the RT-PCR process.

Xpert Xpress CoV-2/Flu/RSV plus (Xpert) contains primers and probes for the amplification and detection of the target genes of N2, E, RdRP (SARS-CoV-2), M, PB2, PA (Flu A), M and NS (Flu B) and genes for the nucleocapsid A and B (RSV). The test applies 45 cycles, and the entire duration of the test including automated extraction and amplification is 36 min.

Target genes of the Savanna RVP4 plus (RVP4) are nsp13 and E (SARS-CoV-2), M1 and M2 (Flu A), NP (Flu B), and NS2 (RSV). The test applies 45 cycles with a duration of 18 seconds per cycle (about 12 min), and the entire duration of the test including automated extraction (about 8 min) and amplification is 20 min.

Both tests were performed according to the instructions for use of the manufacturers, using sample volumes of 300 µL (Xpert) and 250 µL (RVP4), respectively. In case of multiple positive targets for one virus, the lowest Ct-value was chosen when comparing Xpert and RVP4.

## Calculation of costs

Costs were calculated using consumables (list prize of the assays) and work time (hands-on time × salary of technician). As an average salary of a technician, we used the salary scale of laboratory technicians in the public service in Germany (4.250 €/month). The salary was calculated to the second by assuming 21.75 working days per month and a working day of 8 h (9). Invalid runs were not included in the cost calculation. Cost calculation in US dollars was not feasible as the RVP4 tests is not yet available for sale in the United States.

## Statistics

To measure the intra- and inter assay precision of Ct-values, we pooled positive samples to have a final volume of 1.5 mL for each parameter. For the intra assay precision, the test was repeated three times in 1 day. For the inter assay precision, we performed two repeats for two additional days (10). To quantify the precision of Ct-values, the coefficient of variation was calculated for the technical replicates as recommended (10, 11). Descriptive test statistics [i.e., sensitivity, specificity, negative predictive value (NPV), and positive predictive value (PPV)] were calculated for RVP4.

Continuous variables (Ct-values, time) were visually inspected for normal distribution. In the absence of normally distributed variables, a non-parametric test (Wilcoxon rank sum) was applied.

Statistics were performed with "R" and the package "epiR" (Version 4.3.3).

## RESULTS

We tested 292 nasopharyngeal swabs during the 2023/24 respiratory season (September 2023–February 2024). After the removal of duplicate samples ($n = 33$) and invalid test runs (Savanna: $n = 28$, Savanna and Xpert: $n = 1$, Xpert: $n = 1$), 229 samples were entered in the final data set. To be valid, the tests had to report valid results for all four targets.

The median age of included patients was 61.4 years (range: 0.2–92.0). The majority was male (64.6%, $n = 148/229$). At the time of sampling, 20.1% (46/229) of patients had symptoms suggestive for a viral respiratory tract infection with fever being predominant (16.6%, $n = 38/229$), followed by rhinitis (3.9%, $n = 9/229$) and pneumonia (3.5%, $n = 8/229$). The majority did not report signs or symptoms of a respiratory tract infection (79.9%, $n = 183/229$).

According to the reference standard (Xpert), 22.7% ($n = 52/229$) were positive for SARS-CoV-2, followed by Flu A (14.0%, 32/229) and RSV (9.6%, 22/229, Table 1). None of the samples were tested positive for Flu B.

The concordance between Xpert and RVP4 was 96.5% (SARS-CoV-2), 98.7% (Flu A), and 99.6% (RSV). The accuracy of RVP4 was dependent on the target and ranged between 85%–95% (sensitivity), 100% (specificity), 96%–100% (NPV), and 100% (PPV, Table 1).

The median Ct-values were significantly higher in the RVP4 test compared to Xpert for the detection of Flu A. For the other targets, Ct-values were comparable between both tests (Table 2).

For SARS-CoV-2, the median Xpert Ct-values for "Xpert-positive/RVP4-negative" samples ($n = 8$) were higher compared to "Xpert-positive/RVP4-positive" samples ($n = 44$, 39.5 vs. 25.1).

For Flu A, the median Xpert Ct-values for "Xpert-positive/RVP4-negative" samples ($n = 3$) were higher compared to "Xpert-positive/RVP4-positive" samples ($n = 29$, 35.4 vs 23.0).

**TABLE 1** Diagnostic accuracy of the Savanna RVP4 plus (QuidelOrtho) for the detection of SARS-CoV-2, Flu A, and RSV in nasopharyngeal swabs, Germany, 2023–2024

| RVP4 | | Xpert | | Total | Sensitivity (95% CI) | Specificity (95% CI) | NPV (95% CI) | PPV (95% CI) |
|---|---|---|---|---|---|---|---|---|
| | | Positive | Negative | | | | | |
| SARS-CoV-2 | Positive | 44 | 0 | 44 | 85% (72%–93%) | 100% (98%–100%) | 96% (92%–98%) | 100% (92%–100%) |
| | Negative | 8 | 177 | 185 | | | | |
| | Total | 52 | 177 | 229 | | | | |
| Flu A | Positive | 29 | 0 | 29 | 91% (75%–98%) | 100% (98%–100%) | 98% (69%–100%) | 100% (88%–100%) |
| | Negative | 3 | 197 | 200 | | | | |
| | Total | 32 | 197 | 229 | | | | |
| RSV | Positive | 21 | 0 | 21 | 95% (77%–100%) | 100% (98%–100%) | 100% (97%–100%) | 100% (84%–100%) |
| | Negative | 1 | 207 | 208 | | | | |
| | Total | 22 | 207 | 229 | | | | |

For RSV, the Xpert Ct-values for "Xpert-positive/RVP4-negative" sample ($n = 1$) was higher compared to the median "Xpert-positive/RVP4-positive" samples ($n = 21$, 34.8 vs 25.7).

A total of six samples were pooled to achieve a sufficient volume (1.5 mL) to assess the intra- and inter assay precision of Ct-values. For precision, we calculated the coefficient of variation. The intra assay precision of Ct-values was 5% (Flu A), 3% (SARS-CoV-2), and 0% (RSV). Similarly, the inter assay precision was 4% (Flu A), 2% (SARS-CoV-2), and 3% (RSV, Table S1).

The cost per test (staff costs and consumables) was lower in the RVP4 compared to the Xpert (35.78 € vs 52.80 €, Table 2).

## DISCUSSION

We compared the Savanna RVP4 plus with Xpert Xpress CoV-2/Flu/RSV plus and found a high concordance for the detection of SARS-CoV-2, Flu A, and RSV.

The proportion of erroneous and invalid RVP4 tests (9.9%, 29/292) was higher compared to Xpert tests (0.7%, 2/292). This would limit the use of RVP4 as a point of care test, particularly in emergency units, when test runs are not continuously monitored by the staff to re-start the test in case of errors. The error rate of RVP4 was mostly related to specific manufacturing batches of the test cassette, and the manufacturer has identified and solved this problem for the batches used in this study (personal communication with the manufacturer).

Our study also included asymptomatic patients. However, the proportion of positive samples was comparable to a similar diagnostic accuracy study with symptomatic patients for SARS-CoV-2 (19.2% vs 17.0%) or Flu A (12.7% vs 12.3%) (12). None of the samples were tested positive for Flu B which is in line with a very low proportion of Flu B during the study period (September 2023–February 2024). The proportion of Flu

**TABLE 2** Comparison of the Savanna RVP4 plus and Xpert Xpress CoV-2/Flu/RSV plus assay[c]

| | | Savanna RVP4 plus | Xpert Xpress CoV-2/Flu/RSV plus | P-value |
|---|---|---|---|---|
| Median Ct-value (range) | SARS-CoV-2 | 26.0 (15–39) | 28.7 (14.6–44.7) | 0.41 |
| | Flu A | 28.0 (15.1–40-0) | 23.4 (15.4–39.6) | 0.007 |
| | RSV | 28.0 (19.0–40.0) | 25.7 (17.7–35.5) | 0.88 |
| Cost assessment | Cost per test[a] | 33.74 € | 50.80 € | NA |
| | Hands-on time | 300 s | 300 s | NA |
| | Total cost per test[b] | 35.78 € | 52.80 € | NA |

[a]List price, excluding taxes.
[b]Hand-on time × salary/s + cost per test (invalid tests were not included in the cost analysis).
[c]NB: NA, not applicable.

B among all cases of influenza increased only after the study period and became even predominant in week 10 or 2024 (13).

Another frequently used point-of-care test for the detection of SARS-CoV-2 is the ID NOW COVID-19 assay (Abott, Chicago, Illinois, USA), which is an isothermal amplification test. This method has been shown in a Cochrane systematic review to have a lower sensitivity compared to the Xpert Xpress (Cepheid, 76.8% vs 99.4%), while the specificity is comparable between the two tests (99.6% vs 96.8%) using RT-PCR as a reference standard (14). Thus, the sensitivity for the detection of SARS-CoV-2 was higher for the RVP4 (85%, Table 1) compared to ID NOW COVID-19.

Other point-of-care tests for the diagnosis of Flu A/B are the ID NOW Influenza A & B 2 assay (Abbott) or Cobas LIAT Influenza A/B test (Roche, Rotkreuz, Switzerland). A recent study published by Kanwar and colleagues investigated the analytical performance of the ID NOW and Cobas LIAT for the detection of Flu on 201 specimens from a pediatric population. In this study, the ID NOW test showed a similar sensitivity (93.2%) and specificity (99.2%) for the detection of Flu A as the RVP4 in our study (91% and 100%, respectively, Table 1), whereas the sensitivity of the Cobas LIAT for the detection of Flu A reached 100% (Specificity: 99.2%) (15). However, the data published by Kanwar and colleagues are not completely comparable with the data of our study because they included only children (our study: all patients) and all patients had symptoms suggestive for influenza (our cohort: 20.1%). In another study conducted on 744 specimens from adult and pediatric populations, the Cobas LIAT point-of-care test had a sensitivity of 100% and a specificity of 98.1% for Flu A (16).

In our study, the RVP4 had a sensitivity and specificity of 95% and 100%, respectively, for the detection of RSV (Table 1). A similar sensitivity and specificity or the detection of RSV was achieved by Cobas LIAT Influenza A/B and RSV test (Roche, 100% and 99.4%) (16). In summary, a comparison of nucleic acid amplification-based point-of-care tests shows that the RVP4 has similar sensitivities and specificities for the detection of SARS-CoV-2, Flu A, and RSV.

A total of 12 samples yielded discrepant results for SARS-CoV-2, Flu A, and RSV (Table 1). In all discrepant cases, RVP4 was negative and the Xpert test was positive. The higher median Ct-value of the Xpert test in the discrepant results compared to concordant results (39.5 vs 25.1 for SARS CoV-2, 35.4 vs 23.0 for Flu A, 34.8 vs 25.7 for RSV) is indicative for a lower viral concentration, which was not detected by RVP4. This is in line with a higher limit of detection of the RVP4 assay compared to Xpert for SARS-CoV-2 (1,000–3,000 copies/mL vs 131 copies/mL) according to the manufacturers. Similarly, the tissue culture infection dose 50 ($TCID_{50}$) is higher in the RVP4 compared to Xpert for Flu A (42–47 vs 0.004–0.087 $TCID_{50}$/mL) and RSV (0.42–0.50 vs 0.33–0.37 $TCID_{50}$/mL) and, therefore, explains the discrepant results in the detection of Flu A and RSV (Table 1). In addition, all cases with discrepant results for SARS-CoV-2 and Flu A were asymptomatic, suggesting that RVP4 has it strengths to detect clinically relevant cases.

The intra and inter assay precision of our samples was 0%–5% and 2%–4%, respectively, and comparable to a similar study investigating Ct-values for SARS-CoV-2 (6.5% and 2.2%, respectively) (17).

The costs per test are lower for the RVP4 compared to Xpert (Table 2) mostly due to a lower price for the test kit. Considering a comparable test accuracy, the RVP4 can be considered an adequate alternative for the testing for SARS-CoV-2, Flu A, and RSV.

Our study has limitations. First, the study population consisted of a high number of asymptomatic patients. This reflects the infection prevention and control measures in the aftermath of the SARS-CoV-2 pandemic. Since the NPV and PPV depend on the pre-test likelihood, our results should be interpreted with caution if only symptomatic patients are tested. Second, we did not apply a third method to resolve discrepant results.

In conclusion, the Savanna RVP4 has a favorable diagnostic accuracy for the detection of SARS-CoV-2, Flu A, and RSV.

## ACKNOWLEDGMENTS

We thank all technicians of the Institute of Clinical Virology for excellent technical assistance.

The study was supported by institutional funds. We acknowledge the support from the Open Access Publication Fund of the University of Münster. The test platform (Savanna) and test cartridges (RVP4 plus) were provided by QuidelOrtho free of charge.

F.S. designed this study. F.S. and B.K. contributed in methodology and data analysis. B.K. performed the experiments, B.K. and F.S. wrote the original manuscript text and reviewed and edited the writing. Both authors have read and agreed to the published version of the manuscript.

## AUTHOR AFFILIATION

[1]Institute of Medical Microbiology, University Hospital Münster, Münster, Germany

## AUTHOR ORCIDs

Frieder Schaumburg  http://orcid.org/0000-0002-9168-9290

## AUTHOR CONTRIBUTIONS

Büsra Köse, Formal analysis, Methodology, Writing – original draft, Writing – review and editing | Frieder Schaumburg, Conceptualization, Writing – original draft, Writing – review and editing

## ETHICAL APPROVAL

Ethical approval was obtained from the ethical committee of the University of Münster (Ethik-Kommission Westfalen-Lippe, 2023-533-f-S). A written informed consent from patients was waived by the ethical committee.

## ADDITIONAL FILES

The following material is available online.

### Supplemental Material

**Table S1 (Spectrum01153-24-s0001.docx).** Ct values for the calculation of the inter- and intra-assay precision.

### Open Peer Review

**PEER REVIEW HISTORY (review-history.pdf).** An accounting of the reviewer comments and feedback.

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
