## [Reviewer comments · Microbiology Spectrum]

Microbiology Spectrum

Diagnostic accuracy of Savanna RVP4 (QuidelOrtho) for the detection of Influenza A virus, RSV and SARS-CoV-2

Büsra Köse and Frieder Schaumburg

Corresponding Author(s): Frieder Schaumburg, Westfälische Wilhelms-Universität Münster

Review Timeline:

Submission Date:	May 17, 2024
Editorial Decision:	June 7, 2024
Revision Received:	June 11, 2024
Accepted:	June 21, 2024

Editor: Melissa Gitman

Reviewer(s): The reviewers have opted to remain anonymous.

Transaction Report:

DOI: <https://doi.org/10.1128/spectrum.01153-24>

Re: Spectrum01153-24 (Diagnostic accuracy of Savanna RVP4 (QuidelOrtho) for the detection of Influenza A virus, RSV and SARS-CoV-2)

Dear Dr. Frieder Schaumburg:

Thank you for the privilege of reviewing your work. Below you will find my comments, instructions from the Spectrum editorial office, and the reviewer comments.

Major comments:

As the authors correctly state in the limitations section, this study does not have a discriminator test. Potential options could have included repeat test of any discordant results on the same platform to see if the results were reproducible or use a tie breaker method. Is there still any stored sample available to perform such testing as this would greatly strengthen the manuscript.

Minor comments:

Lines 113-4 (we compared....) - It is clear 1 swab was collected per patient, please clarify whether the Xpert and the RVP4 were run in parallel at the time of receipt or whether the RVP4 swabs were tested at a later time. It is important to explicitly state this, to show if that pre-analytical variables were consistent between tests.

Line 179 - the sentence reads for RSV the median Xpert Ct-values.. however, only one sample is included. Median is the incorrect term here. Only a descriptive comparison can be made here.

line 183- 185 - It would be helpful to elaborate on what values are being provided here for precision. One can infer that given in the methods there is mention of the calculation of CoV but it would be helpful if this was explicitly stated in this paragraph. Table 1 - One sample with RSV positive by Savanna RVP4 is negative by Xpert. Can you provide the CT value of this sample?

Revision Guidelines

Sincerely,
Melissa Gitman
Editor
Microbiology Spectrum

Universitätsklinikum Münster . 48129 Münster . [42800]

To
Microbiology Spectrum
Editorial office

Universitätsklinikum Münster
Institut für Medizinische Mikrobiologie
Univ.-Prof. Dr. med. Frieder Schaumburg
Direktor

Domagkstraße 10
48149 Münster
www.ukm-lageplan.de

T +49 251 83-52767
Servicezentrale: T +49 251 83-55555

frieder.schaumburg@ukmuenster.de
www.ukm.de

Münster, 21.06.2024

Spectrum01153-24: Diagnostic accuracy of Savanna RVP4 (QuidelOrtho) for the detection of Influenza A virus, RSV and SARS-CoV-2

Dear Dr. Gitman,

we thank you and the reviewers for the time to evaluate our work, we particularly appreciate the timely process which is not common nowadays. Please find enclosed how we addressed the concerns and suggestions in point-by-point reply.

Major comments

As the authors correctly state in the limitations section, this study does not have a discriminator test. Potential options could have included repeat test of any discordant results on the same platform to see if the results were reproducible or use a tie breaker method. Is there still any stored sample available to perform such testing as this would greatly strengthen the manuscript.

Reply: We agree that this is a relevant limitations. Unfortunately, samples are no longer available as they were either completely used as part of the tests in this study or they have been used for the pooled samples for intra- and inter assay precision. We tried to address the discrepant results by looking at the Ct values and were able to show that Ct values were usually higher in the Cepheid test if positive results were not confirmed by the RVP4 test suggesting a higher level of detection of RVP4.

Minor comments:

Lines 113-4 (we compared....) - It is clear 1 swab was collected per patient, please clarify whether the Xpert and the RVP4 were run in parallel at the time of receipt or whether the RVP4 swabs were tested at a later time. It is important to explicitly state this, to show if that pre-analytical variables were consistent between tests.

Reply: Thank you, we now clarified this issue in the revised version (line 103-104).

Line 179 - the sentence reads for RSV the median Xpert Ct-values.. however, only one sample is included. Median is the incorrect term here. Only a descriptive comparison can be made here.

Reply: We agree and changed the sentence accordingly (line 180-181).

Line 183- 185 - It would be helpful to elaborate on what values are being provided here for precision. One can infer that given in the methods there is mention of the calculation of CoV but it would be helpful if this was explicitly stated in this paragraph.

Reply: We now provide the Ct values in the supplement that were used to calculate the precision. We now also state that precision was calculated as the coefficient of variation (CoV).

Table 1 - One sample with RSV positive by Savanna RVP4 is negative by Xpert. Can you provide the CT value of this sample?

Reply: Thank you for pointing out this issue, as we identified one mistake in our table, which is now corrected in the table, abstract and text. There is no RSV Savanna +/Xpert – sample in our dataset. In addition, we thoroughly checked all numbers again in our manuscript for consistency within the text and with the table and corrected these numbers if applicable.

We thank you for your time in this matter,

Frieder Schaumburg

Re: Spectrum01153-24R1 (Diagnostic accuracy of Savanna RVP4 (QuidelOrtho) for the detection of Influenza A virus, RSV and SARS-CoV-2)

Dear Dr. Frieder Schaumburg:

Your manuscript has been accepted, and I am forwarding it to the ASM production staff for publication. Your paper will first be checked to make sure all elements meet the technical requirements. ASM staff will contact you if anything needs to be revised before copyediting and production can begin. Otherwise, you will be notified when your proofs are ready to be viewed.

Sincerely,
Melissa Gitman
Editor
Microbiology Spectrum